

# Uncertainties in Coastal Flood Risk Assessments in Small Island Developing States

Matteo U. Parodi [1], Alessio Giardino [1], Ap van Dongeren[1], Stuart G. Pearson[1,2], Jeremy D. Bricker[2], Ad J.H.M. Reniers[2]

[1] Deltares, Unit Marine and Coastal Systems, Boussinesweg 1, 2629 HV Delft, The Netherlands
[2] Civil Engineering and Geoscience, Delft University of Technology, 2628 CN Delft, The Netherlands

*Correspondence to*: Matteo U. Parodi (parodi.mat@gmail.com)

**Abstract.**

Considering the likely increase of coastal flooding in Small Island Developing States (SIDS), coastal managers at the local
and global level have been developing initiatives aimed at implementing Disaster Risk Reduction (DRR) measures and adapting to climate change. Developing science-based adaptation policies requires accurate coastal flood risk (CFR) assessments, which are often subject to the scarcity of sufficiently accurate input data for insular states. We analysed the impact of uncertain inputs on coastal flood damage estimates, considering: (i) significant wave height, (ii) storm surge level and (iii) sea level rise (SLR) contributions to extreme sea levels, as well as the error-driven uncertainty in (iv) bathymetric and (v)
topographic datasets, (vi) damage models and (vii) socioeconomic changes. The methodology was tested through a sensitivity analysis using an ensemble of hydrodynamic models (XBeach and SFINCS) coupled with an impact model (Delft-FIAT) for a case study at the islands of São Tomé and Príncipe. Model results indicate that for the current time horizon, depth damage functions (DDF) and digital elevation model (DEM) dominate the overall damage estimation uncertainty. We find that, when introducing climate and socioeconomic uncertainties to the analysis, SLR projections become the most relevant input for the
year 2100 (followed by DEM and DDF). In general, the scarcity of reliable input data leads to considerable predictive error in CFR assessments in SIDS. The findings of this research can help to prioritise the allocation of limited resources towards the acquisitions of the most relevant input data for reliable impact estimation.

## 1 Introduction

Small Island Developing States (SIDS) are increasingly under threat of coastal flooding, which challenges the safety and
sustainability of their society and the growth of their economies (OECD World Bank, 2016). The consequences that they will face due to climate change, particularly considering the risk of coastal floods, are disproportionate to their intrinsic resilience. For example, sea level rise (SLR) will exacerbate the impacts and frequency of coastal hazards at many islands around the world (Storlazzi et al., 2018; UN-OHRLLS, 2015). This situation has recently led to initiatives aiming to increase the resilience of insular communities by using robust coastal flood risk (CFR) assessments as a necessary first step to develop sustainable
adaptation strategies.



Existing hydrodynamic models can achieve satisfactory levels of accuracy in estimating flood hazards, particularly at the local scale (Bertin et al., 2014; Dresback et al., 2013; Giardino et al., 2018; Monioudi et al., 2018; Storlazzi et al., 2018). Nevertheless, CFR assessments are subject to a wide range of errors and uncertainties, divided into *aleatory uncertainties*, i.e. related to the intrinsic randomness of reality, and *epistemic uncertainties*, due to imperfect knowledge and lack of data (Uusitalo et al., 2015). High-quality input data is often particularly scarce for many of these islands, due to their remoteness and limited economic resources. This forces risk modelers to make additional simplifying assumptions.

For the assessment of coastal hazards and risk, datasets covering the entire globe in low resolution must often be used in the absence of detailed local data. These global datasets are often inaccurate, which negatively affect the trustworthiness of the model and ultimately the outcome of the study. Cook and Merwade (2009), Kulp and Strauss (2019) and Van de Sande et al. (2012) have acknowledged the unreliability of publicly available digital elevation models (DEMs) to represent the exposure to coastal floods, while Cea and French (2012), Hare et al. (2011) and Plant et al. (2002) have highlighted the significant uncertainty that low-resolution bathymetric datasets bring into coastal hazard modeling. Global bathymetric datasets (e.g. GEBCO) lack information on nearshore depth, especially over reefs or in bays, while global topographic datasets (e.g. SRTM, ASTER) experience contamination of terrain elevation data due to buildings, vegetation canopies, and other objects that are averaged into the elevation representing each coarse pixel of the dataset. However, considerable efforts are being directed to improve the quality of satellite-derived DEMs. Very recently, improved global datasets such as MERIT (Yamazaki et al., 2017)and CoastalDEM (Kulp and Strauss, 2019) have been published, which correct for vegetation and building elevation biases. In the near future, data from satellite missions like ICESAT 2 will produce even more accurate DEMs.

The damage assessment represents a step of a CFR analysis severely affected by both the paucity of reliable damage information (Apel et al., 2004; Merz and Thieken, 2009; De Moel and Aerts, 2011; Prahl et al., 2016; Wagenaar et al., 2016), and the simplifications that are necessary to quantify the vulnerability of human and natural assets. Furthermore, the uncertainty of damage modeling is exacerbated in data-poor SIDS, where accurate data and models are lacking, requiring strong assumptions. Indeed, often damage curves are taken from literature and applied in different areas making few, if any, adjustments (Schroter et al., 2014; Wagenaar et al., 2016). Moreover, extreme sea level (ESL) events constitute a considerable portion of the uncertainties in a CFR analysis, as their statistical estimation method is based on extrapolating from limited duration of recorded data and requires the choice of a probability distribution function (pdf) (Wahl et al., 2017).

Finally, to develop long term adaptation plans, future risk estimates including changes to human and natural systems are required, which introduce further assumptions and uncertainty. Indeed, both future climatological and societal changes can significantly impact the model outcome, and disregarding them may lead to poor coastal zone planning and underestimation of future damages (Bouwer, 2013; Bouwer et al., 2010).

Several studies have attempted to quantify the uncertainty in flood risk estimates, both for coastal (Hinkel et al., 2014; De Moel et al., 2012; Vousdoukas et al., 2018) and riverine floods (Apel et al., 2004; Egorova et al., 2008). Vousdoukas et al. (2018) and De Moel et al. (2012) performed an uncertainty analysis on CFR assessments for two case studies in Europe,





indicating the quality of coastal protection information and the shape of the depth damage functions (DDFs) as the most influential input for flood damage estimates uncertainty, respectively.

A quantification of the relative contribution of the uncertainty sources has yet not been conducted for SIDS, where the scarcity of input data exacerbates the model outcome error and uncertainty. We therefore present a method to directly compare the

relative importance of uncertainty sources on the estimation of coastal flood damages, extending the analysis to present-day and future risk predictions by modeling future damages for the years 2050, 2070 and 2100. For this purpose, this study describes a developed framework that examines different uncertainty sources, including the components contributing to ESLs, namely (i) waves, (ii) tides and (iii) SLR projections; (iv) bathymetry dataset; (v) DEM; (vi) damage models and (vii) socioeconomic growth.

## 2 Case Study

The methodology was applied to two coastal villages in the Democratic Republic of São Tomé and Príncipe, an archipelago that comprises two main islands and several islets, located in the Guinea Gulf (Fig. 1a). The two villages were selected based on their high vulnerability to coastal flooding hazards and on the availability of local information to conduct the CFR assessment. The small size and location of the islands, in combination with their colonial history, have significantly hampered

their economic development, increasing their susceptibility to natural disasters and hindering a sustainable future for the communities (Deltares and CDR, 2019). The village of Pantufo is situated on the north east side of the island of São Tomé (Fig. 1b), bordered by a partially sandy and rocky beach (Fig. 1c). The village of Praia Abade is located on a north-east side of the island of Príncipe, at the southern end of a bay (Fig. 1b) and it is bordered by a sandy pocket beach (Fig. 1d). Fishing represents the main economic activity for both villages. Near the coastline, houses are often made of wood or poor-quality

concrete and elevated to prevent flooding damages.

Both communities are on the lee side of the islands sheltered from the largest and most frequent southerly swell waves. The area is off the track of most tropical cyclones and storms, experiencing a calmer wave climate than other extratropical regions (Alves, 2006). However, these communities are still prone to hazardous rainfall and coastal flooding from occasional big southerly swells, which damage buildings and fishing boats (Deltares and CDR, 2019).

## 25  3 Data and Methods

### 3.1 Modeling Approach

Coastal risks at the two villages were estimated using a chain of models and data as shown in Fig. 2. Most of the input data are characterised by uncertainties that contribute to uncertainty in the final damage estimates. The major sources of uncertainty considered in this study are highlighted with red boxes in Fig. 2. Risk is computed as a combination of hazard, exposure and

vulnerability (Kron, 2005). Specifically, "hazard" is the probability and magnitude of an event with negative impacts.



"Exposure" means the assets that are exposed to the hazard, and "vulnerability" refers to the damage inflicted upon the exposed asset, under a specific hazard. These components were modeled separately and are shown respectively in orange, green and yellow boxes in Fig. 2.

Coastal floods are driven by nearshore ESLs. Flood maps representing the coastal flood hazards were computed using a 2-D
SFINCS (Leijnse et al. 2020, submitted) model, based on the peak six hours of a 24-hour reference storm and with land surface elevation derived from a DEM (Fig. 2). To describe the storm, a storm surge water level was imposed over a spring tidal water level, and offshore waves are explicitly included (Fig. 3). SFINCS is a computationally-efficient coastal zone flood model, and covered the area of interest with a rectilinear grid and a spacing of 5 m.

SFINCS is forced with water levels computed using the nonhydrostatic version of the model of XBeach (De Ridder et al.,
2019 under review; Roelvink et al., 2009, 2018; Smit et al., 2010), taken at 2 m water depths from cross-shore transects (Fig. 2). These transects had a minimum grid size of 1 m, running from approximately 20 m water depth offshore to an inland elevation of approximately 10 m. The nearshore boundary conditions for XBeach are computed using transformation matrixes in the DELFT3D-WAVE (SWAN) model (Booij et al., 1994) as described in Deltares and CDR (2019).

The damage assessment was conducted through the model Delft-FIAT (Slager et al., 2016). FIAT (Flood Impact Assessment
Tool) is a flexible Open Source toolset, where direct damages are estimated at the unit level (e.g. a single building or piece of infrastructure). Combining information on the exposed assets, DDFs, and flood maps, expected damages from single events were obtained (Fig. 2).

DDFs define, for each asset type, the relation between a given flood depth and the consequent direct damages (Messner and Meyer, 2005) and are widely used in flood damage modeling due to their simplicity (Schroter et al., 2014). In this analysis,
arbitrarily chosen 100-year return period extreme sea levels event were modeled. The analysis focused on direct and tangible damages to boats and buildings, which dominate the total flood impacts over the communities.

### 3.2 Data

Multiple data sources were used as a basis to perform the CFR analysis. Table 1 contains an overview of the uncertainty sources investigated, indicating their baseline values and the variations from it, for each different input variable investigated.
The baseline scenario uses a combination of the best available input data (i.e. highest resolution or value in which we have the highest confidence). To estimate the uncertainty in our CFR analysis, we tested variations from this scenario using alternative available data sources or high and low percentiles of a given probability distribution. Each input variable and related source of uncertainty for present and future conditions is described in the next two sections.






### 3.2.1 Present scenario

*Significant wave height*

The ERA-Interim dataset (Dee et al., 2011) by ECMWF (European Centre for Medium-Range Weather Forecast) was used. The dataset provides 6-hourly significant wave height ($H_s$) data over more than 30 years and was used to estimate the 100-year

return period event for $H_s$, conducting an extreme value analysis (EVA). A peak over threshold technique (Caires, 2011) was conducted on the nearshore wave conditions, fitting a Generalised Pareto Distribution (GPD) (Pickands, 1975) to the peaks of clustered excesses over a threshold. The 98th percentile of the $H_s$ distribution was selected as threshold, as recommended by Wahl et al. (2017).

Commonly, extreme hydrodynamic boundary conditions are represented with probability distributions. However, these

distributions are fit to measured data and attempt to estimate values for return periods longer than the length of the available data, thus already introducing uncertainty in the model. Therefore, the uncertainty was taken into account by using the 5th, 50th and 95th percentile values of $H_s$ (Table 2) in XBeach (Fig. 2).

*Storm surge*

The estimation of storm surge levels was based on the dataset by Muis et al. (2016), a global water level reanalysis based on daily maxima over 35 years. In an identical manner to $H_s$, the 5th, 50th and 95th percentile values for the 100-year storm surge level were estimated (Table 2), aiming to reproduce its uncertainty. The probability distribution of storm surge and significant wave height were assumed to be independent of each other, therefore without making use of a joint probability distribution.

*Bathymetry*

Bathymetry controls the wave transformation mechanisms and ultimately the flooding on land. To explore the role of bathymetry data uncertainty, two datasets were used. The General Bathymetric Chart of the Oceans (GEBCO) (Weatherall et al., 2015), a publicly available bathymetric dataset, was compared to a locally collected dataset (Deltares and CDR, 2019). GEBCO has a coarser horizontal resolution than the local dataset (approximately 900 m and 50-100 m, respectively). Using

bathymetry data points with coarse resolutions to generate a digital seabed introduces several errors and uncertainty, due to the unresolved terrain variability between measured points (Hare et al., 2011; Plant et al., 2002). The local measurements were taken during a campaign in December 2018, when cross-shore transect profiles were collected at the two communities, using a handheld echo sounder (Deltares and CDR, 2019).

*Digital elevation model*

Digital elevation models are numerical representations of the earth surface elevation. Similar to bathymetric datasets, DEMs with lower resolution will introduce more uncertainty, due to interpolation errors. Furthermore, systematic errors that stem from a bias in the elevation values are often included in the datasets and have a considerable impact on flood risk estimates



(Cook and Merwade, 2009; Kulp and Strauss, 2019; Paprotny et al., 2019; Van de Sande et al., 2012). Indeed, global and satellite-derived DEMs often have a low vertical accuracy for CFR assessments, being surface models where terrain elevation values may be overestimated due to land cover (e.g., tree canopies and the built environment).

During the site campaign, topography information was derived from Unmanned Aerial Vehicle (UAV) imagery (Deltares and CDR, 2019), using the Drone2Map software from ESRI. UAV-derived DEMs have been proven to show higher vertical accuracy than satellite-derived DEMs (Gonçalves and Henriques, 2015; Hashemi-Beni et al., 2018; Leitão et al., 2016). The UAV measurements were horizontally and vertically referenced using one Ground Control point in EGS 1984 ellipsoid vertical datum. In order to quantify the effect of the DEM vertical accuracy on flood estimates, multiple globally available, satellite-derived datasets were collected and compared against the UAV-derived DEM. The latter, with a horizontal resolution of roughly 10 cm, was assumed to have the highest vertical accuracy. The investigated satellite-derived DEMs include TanDEM-X, TerraSAR-X, MERIT, ASTER and SRTM. Their horizontal resolution and vertical accuracy are described in Table 3.

*Depth damage function*

Depth damage functions (DDFs) describe the vulnerability of the assets at risk in the event of a flood, relating a given flood depth to a damage factor that indicates the percentage of the lost asset value. DDFs span a large variety of flooding types and building strengths, allowing for the computation of different damage scenarios (Schroter et al., 2014). However, numerous simplifications are introduced in designing such curves, such as fitting them to sparse data values and often disregarding important processes like wave forces and flooding duration. This is partly due to the high complexity of damage physics (which still lack thorough understanding) and to the scarcity of buildings information (Apel et al., 2004; Merz et al., 2007; Merz and Thieken, 2009; Wagenaar et al., 2016). In SIDS, locally-derived DDFs are rarely available, forcing risk modelers to apply DDFs originally derived for different geographic areas and flood types. The uncertainty of this input was represented by using a variety of possible shapes and types used in CFR analyses retrieved from literature. Figure 4 and Table 4 contain a summary of the DDFs used. Generally, two main types of DDFs are used. Convex curves are representative of more flood-resilient assets that only undergo significant damages at high flood depths (e.g. American Samoa and DSM curves in Fig. 4). On the other hand, a concave shape represents less flood-resilient building, undergoing significant damages already at small flood depths (e.g. Sint Maarten or JRC curves in Fig. 4). Concave-shaped DDFs may be preferable at most SIDS, being most representatives of buildings in developing countries. Economic values for different building types were collected during site visits (Deltares and CDR, 2019). In this research, a single economic value was used to represent an average building in each community.





### 3.2.2 Future scenarios

To perform future risk analyses, changes in the drivers and receptors of risks must be accounted for; thus, climatic changes and socioeconomic development were included in the study. To account for future climatic changes, only SLR was included. Other processes, such as astronomical tides, storm surge levels, wave heights and local morphology were assumed to be

constant in time. This is consistent with other uncertainty studies (Hinkel et al., 2014; Vousdoukas et al., 2018), where only the mean sea level was assumed to be affected by climate change.

*Sea level rise scenario*

The dataset of global probabilistic projection of sea levels under the Representative Concentration Pathway (RCP) 8.5

scenario, developed by the Joint Research Centre (JRC) (Vousdoukas et al., 2016) was used. The choice of RCP 8.5 relies on the fact that the 90% confidence interval of this projection also captures most of the 90% confidence interval values under the RCP 4.5 scenario. In our approach, SLR scenarios were used to increase the static water level (Fig. 2). However, the range of future SLR remains uncertain, considering the variability of the numerous processes that affect it. Therefore, choosing a single SLR scenario limits the understanding of the system susceptibility to future flood risk and hides the uncertainty in the

prediction. To reproduce this uncertainty, the $5^{th}$, $50^{th}$ and $95^{th}$ percentiles values of SLR projections for the study area, were simulated (Table 5), for the three future time horizons 2050,2070, and 2100.

*Socioeconomic scenario*

For the case of São Tomé and Príncipe, urbanization and global development trends drive an increase in the number and

value of exposed assets in coastal communities, for both mid- and long- term time horizons (2050, 2070 and 2100) (Deltares and CDR, 2019). Riahi et al. (2017), in collaboration with the IPCC panel, have developed a set of possible societal developments, Shared Socioeconomic Pathways (SSPs), which vary according to the efforts adopted to mitigate and adapt to climate change pressures. They are designed to span a wide range of uncertainty in future human developments and define future economic variables, such as the gross domestic product (GDP). GDP and population growth rate were used in this

analysis as a proxy to compute future asset values as follows:

$$Asset\ Value\ Growth = \frac{GDP_{t,s}}{GDP_{2018}} \cdot \frac{Population_t}{Population_{2018}} \tag{1.1}$$

where $GDP_{t,s}$ is the GDP at the year $t$, under the SSP $s$. Three SSPs (SSP 2, 3 and 4) were considered to reproduce

socioeconomic growth uncertainty, as they cover the largest range of GDP growth values for the year 2100 (Fig. 5).



### 3.3 Baseline scenario and variations

A sensitivity analysis was conducted, testing all the possible input values combinations in the model train, changing multiple inputs at once. This resulted in a total of 35,280 simulations. We considered the following scenario as "baseline": offshore ESLs described by the 50th percentile of storm surge, $H_s$ and SLR, the locally measured bathymetry, the DEM derived by UAV

aerial imagery, the DDF developed for São Tomé and Príncipe and the "business as usual" SSP 3. For each input and simulation, the ratio of change of the damage estimate from the simulation with the baseline value for that input was computed. Values higher and lower than one express, respectively, an over- and under-estimation of the damages, while the range of values expresses the introduced uncertainty around each input parameter, as summarised in Table 1.

### 4 Results

The computed flood maps for Praia Abade and Pantufo for the baseline scenario are shown in Fig. 6. Praia Abade is more flood-prone than Pantufo, where the coastal topography is steeper and the village on higher ground.

The effect of each input on the estimated damages is represented with box plots in Fig. 7. Each panel represents the ratio between estimated damages for a given scenario compared to the baseline scenario, for different time horizons. Changes in the range of results through time for a particular input may be explained by both a variation of the intrinsic uncertainty of the

input, but also by a change of its sensitivity due to the influence of another input. For example, a change in the terrain slope may alter the sensitivity of flood damages to changes in the storm surge level.

*Hydrodynamic forcing*

As expected, varying the values of $H_s$ and storm surge affects the estimated damages by between 0.5-1.75 and 0.7-1.6 times

the baseline scenario, respectively, in the current time horizon (Fig. 7a, yellow and turquoise boxes). Both their impacts on output uncertainty decrease in time, as can be seen from the decreasing size of the boxes and whiskers in Fig. 7b-c-d. As these inputs are assumed stationary in time, their impact reduction is due to the influence of other inputs to their sensitivity.

*Bathymetry*

Modeling the damages using the coarser bathymetry dataset GEBCO increases the mean damage distribution of 1.25 with respect to the baseline scenario and under the current time horizon (Fig. 7a, blue box). Similarly to storm surge and $H_s$, the impact of bathymetry on the damage estimates decreases in time, with the boxes and whiskers decreasing in size in Fig. 7b-c-d. Figure 8 shows the histograms of damages for the current time horizon (1,260 simulations) using a single bathymetry dataset, highlighting the effect of using one dataset over another.

Comparing the distribution of estimated damages for the current time horizon from all input combinations with a single bathymetry dataset shows an increase in the mean for both locations when GEBCO is used while the width of the 50 percent





confidence interval of results increases by 20 percent. This indicates that the GEBCO profiles are more sensitive to changes in other input conditions than the locally collected profiles.

*Digital elevation model*

Most DEMs highly underestimate flood risk compared to the UAV-derived DEM (Fig. 7a, green box). The global DEMs indicate that almost no flooding will occur, as a result of their low vertical accuracy and positive bias. For both locations, TanDEM-X has the highest vertical accuracy amongst all satellite-derived DEMs with a positive bias of 3.2 m in Praia Abade and 2.9 m in Pantufo (Table 6), although it underestimates the damages (Fig. 9). The impact on damage uncertainty is considerable for all four time-horizons, particularly in 2100, with estimated damages ranging between 0.25 and 2.9 times the

baseline scenario (Fig. 7d, green box). This indicates that the effect of DEMs becomes more sensitive in time to changes in other input conditions. TerraSAR-X is the only DEM underestimating the elevation, explaining the considerable number of upper outliers in the box plots (Fig. 7a-b-c-d green boxes).

Comparing the distribution of estimated damages from all input combinations but using only the UAV-derived vs. TanDEM-X datasets, one can see that the latter results in a considerably smaller mean damage in Pantufo (from 73,000 to 43,000 Euros

(EUR), Fig. 9a) and Praia Abade (from 89,000 to 35,000 EUR, Fig. 9b). Furthermore, the 50 percent confidence interval is reduced. An explanation of the very low reliability of satellite-derived DEMs for our case study might be found in the negative correlation between their vertical accuracy and terrain slope. Indeed, Gorokhovich and Voustianiouk (2006) have found an increase in the prediction error given by SRTM on steeper slopes and mountainous areas, such as the volcanic islands of São Tomé and Príncipe.

*Depth damage function*

The estimated damages show a considerable uncertainty and spread of results depending on the DDF applied. For the current time horizon, DDFs hold the largest impact on model outcome of any input variable, with estimated damages ranging between 0.25 and 4 times the baseline scenario (Fig. 7a, purple box). Their range of uncertainty only slightly decreases through time

(Fig. 7b-c-d). The majority of alternative DDFs are concave and show lower impacts compared to the convex baseline DDF, in particular for low flood depths (Fig. 4), therefore resulting in a box with a mean smaller than one (Fig. 7, green boxes).

*Sea level rise*

SLR initially has a similar impact on the uncertainty of the damages for the year 2050 as $H_s$ and storm surge (Fig. 7b, black box), although this significantly increases for time horizons further in the more distant future. Indeed, sea level rise has the

most considerable spread of results in the year 2100, 0.5-3.7 times the baseline scenario (Fig. 7d, black box). This is partially due to the increasing uncertainty in SLR estimates for the year 2100, as future climate modeling assumptions become weaker for longer time horizons.





*Socioeconomic scenario*

The uncertainty brought by socioeconomic changes is limited in this framework. Indeed, varying the selected SSP does not yield a significant variation of the model outcome, and the highest spread of results is found for the year 2100 (0.6-1.3 times the baseline, Fig. 7d), when uncertainties in the prediction of social development become larger. However, including

socioeconomic factors in the risk estimates increases the economic value of the assets at risk, and thus increases the potential damage. Figure 10 shows the computed damages through time, using three modeling approaches: including only climate change-induced SLR, including only socioeconomic changes, and including both. Future damages are remarkably higher when taking the economic development of the communities into account. When both changes are included the damages of the baseline scenario increase by a factor of 35 in Pantufo and 50 in Praia Abade for the year 2100 (Fig. 10a-b, blue line).

Combining SLR and socioeconomic growth increases the damages non-linearly, as the former increases the hazard and the latter affects the value of exposed assets, therefore acting on different risk components.

### 4.1 Relative Importance

The relative importance of the investigated input variables is depicted in Fig. 11. DDFs and DEMs have the largest relative weight (Fig. 11) for the current time horizon. The relative weights are computed by comparing the width of the range of results (Fig. 7) given by each input through time and scaled to unity. For future risk estimates, the uncertainty coming from SLR continuously increases and it becomes dominant for the year 2100, followed by DDFs and DEMs. Socioeconomic changes

have a somewhat more constrained relative impact, although increasing through time. $H_s$, storm surge, and bathymetry have the smallest relative effect on damage estimates, decreasing with time. However, their impact decreases also absolutely, as their range of results becomes narrower through time (Fig. 7a-b-c-d, yellow, turquoise and blue boxes). Their reduction is linked to the change in mean sea level due to SLR, which leads to the exceedance of thresholds in the elevation that reduce the sensitivity of flood damage estimates to these inputs.

### 5 Discussion

This paper presents an investigation of multiple uncertainty sources in relation to CFR assessment at two small islands, highlighting the consequences of the paucity of reliable input data for SIDS. The results provide a useful indication and highlight the need of collecting higher quality data. Furthermore, the impact of SLR predictions becomes significantly more

important as the time increases, becoming dominant for risk estimates at the end of this century. The baseline scenario is composed by the best available input data (i.e. value in which we have most confidence or with the highest resolution). To





assess the uncertainty in our CFR analysis, we varied this scenario based on alternative available data sources or high and low percentiles of a given probability distribution, estimating the impact brought by each uncertainty source on the damage estimation. However, there are a number of assumptions that go into these estimates, which are discussed below.

### 5.1 Assumptions & Limitations

*Present-day scenario*

We performed a global sensitivity analysis, exploring the whole input space domain. All possible input combinations were tested, leading to the presence of dependencies in the behaviour and response of some inputs. This approach has the advantage of examining input combinations that may include non-linear interactions (Uusitalo et al., 2015). Most notably, $H_s$ and storm surge have experienced a decrease in their impact on damage estimates uncertainty in more distant time horizons (Fig. 7a-b-c-d). This was linked to the action of SLR, which led to a change in the terrain slope at the coastline that altered the sensitivity of damage estimates to changes in storm surge and $H_s$. This demonstrates how uncertainty in one input variable can affect the uncertainty in the estimate brought by another input variable.

The choice of uncertainty sources and their range of values and datasets, although subjective, allows for an indication of the most important uncertainty regarding risk analyses in SIDS. The choice of inputs that were analysed was balanced carefully between comprehensiveness of the analysis and computational expenses. Therefore, a number of factors were discarded, including small scale adaptation measures that are not represented in the DEMs and other sources of flood hazards (e.g. rainfall events). The combination of storm surge peak with the spring-neap tidal variability (Vousdoukas et al., 2018) has also been disregarded, to model a worst-case scenario where the storm peak and spring high tide occur simultaneously (Fig. 3).

We used advanced hydrodynamic models which enabled us to include short wave processes and their effect on floods, an aspect that can lead to intensified flooding consequences (Storlazzi et al., 2018). However, each model contains numerous assumptions and simplification that translate into further uncertainties in the output estimate (Loucks and Van Beek, 2017; Uusitalo et al., 2015). These model uncertainties were disregarded.

Finally, in the damage estimation, only direct and tangible damages were considered, whereas loss of life, natural habitat and other indirect damages were discarded, therefore leading to an underestimation of the total damages.

*Future scenarios*

Societal developments have been implemented in a rather simplistic way, considering only the future asset and population growths. In particular, migration patterns, global economic trends, projected land use, and wealth distribution could be included to further strengthen the methodology. However, the results have still shown the detrimental effect of disregarding socioeconomic changes, responsible for an increase of a factor 50 in the damage estimates (Fig. 10a, blue line).

The spatial distribution of houses built in the future was assumed to be identical to the current one. Since people may relocate to areas with lower flood risk, this assumption could result in an overestimation of the damages. Furthermore, the possibility of exceeding the level of available land for new constructions is not considered, which may have led to estimating an unrealistic





population growth in the communities. Moreover, only one representative type of building was included in the analysis, using a weighted averaging approach based on the distribution of building types. This assumption could yield to an underestimation of flood damages in the case that most of highly valuable buildings are in the most hazard-prone area. Nevertheless, this assumption was supported by the heterogeneous spatial distribution of buildings in Praia Abade and Pantufo. The investigation

of uncertainty in future exposure has not been extensively studied but can have a strong impact on the risk assessment (Bouwer, 2013).

Climate change impacts on future risk predictions were only considered in a limited way, evaluating just the role of SLR. Changes in significant wave height, storm surge and bathymetry were omitted from our analysis, as in other similar studies (Hinkel et al., 2014; Vousdoukas et al., 2018), which leaves their impact on future damage estimates unquantified.

*Applicability of the results to other locations*

The methodology was applied to two coastal communities, respectively on the islands of São Tomé and Príncipe. Although the two villages were located at two different islands, with rather different local geomorphology, the results were rather similar. Therefore, we believe that the general findings from this research could be translated to other SIDS.

## 6 Conclusions

This study aims to better understand uncertainty in coastal flood risk (CFR) in Small Island Developing States (SIDS). The methodology and outcomes were derived based on an assessment of two villages located on the two islands of São Tomé and Príncipe where locally-measured data was available to be compared with publicly-available global datasets. Investigating the

uncertainty propagation from imperfect input data along the whole risk assessment may guide the allocation of limited financial resources to collect the most relevant data more accurately for CFR analyses in SIDS.

The uncertainty investigation was performed using an ensemble of hydrodynamic and impact models, estimating flood damages for a 100-year event. Different input sources of uncertainty were investigated, including (i) waves, (ii) tides, (iii) SLR projections to ESLs, (iv) bathymetry and (v) topography datasets, (vi) damage models (DDFs) and (vii) socioeconomic

changes.

Considerable uncertainty is found in the estimation of flood damages, highlighting the challenges of performing CFR analyses for SIDS. For the current time horizon, the choice of DDF, followed by topography information (DEM), are the main contributors affecting the uncertainty of the output, varying the estimated damages, with a factor ranging between 0.25-4 and 0.3-2.5, respectively relative to the baseline case. For future damage estimates, SLR predictions become the input with the

highest impact on damages estimates. DEM and DDF still carry considerable uncertainty and are ranked second and third in importance (Fig. 11). SLR and especially economic and population growth drive enormous increases in future expected risk, with mean damage estimates of the baseline scenario increasing by up to a factor 50 from the present day. Nevertheless,


socioeconomic changes have a smaller uncertainty compared to other inputs, partially due to their limited model implementation. We thus recommend future research in improving the implementation of socioeconomic changes in risk modeling. $H_s$, storm surge and bathymetry have a more confined impact on the overall damage estimate uncertainty and their relative weight slightly decreases through time.

Using low-quality input data leads to a significant error in the prediction, together with a variation in the level of uncertainty reproduced by the model. This negatively affects the model's trustworthiness, as it may give unwarranted confidence in its output. Complex hydrodynamic models that include multiple physical processes and which can achieve a high level of accuracy in the prediction already exist. However, the efforts put into developing these models can be futile whenever incorrect input data is used, suggesting that the improvement of data-collecting techniques should become a priority. If reducing uncertainty

requires obtaining additional information, then the value of this additional information must exceed the cost of obtaining it. This value will be the reduction of the uncertainty brought by the information.

Furthermore, we recommend focusing on improving DEM quality, collecting damage information, and improving the reliability of SLR projections, as they represent the critical factors affecting the uncertainty in coastal flood damage estimates in SIDS.

**Author contributions:**

MUP, AG, AvD and SGP jointly conceived the study. MUP analysed the data and together with AG prepared the paper, with all the authors discussing results and implications and commenting on the paper at all stages.

**Data Availability:**

This work relied on public data as inputs, which are available from the providers cited in section 3. Locally measured

topographic and bathymetric information made available from Deltares and CDR International.

**Competing interests:**

The authors declare they have no conflict of interest.

**Acknowledgements:**

The authors would like to thank Elena Vandebroek, Luisa Torres Duenas, Tim Leijnse, Kees Nederhoff and Bouke Ottow

from Deltares, Jenny Pronker and Herald Vervoorn from CDR International for their essential contribution in setting up the hazard and risk modelling framework and collecting the local data used in the study. Furthermore, we would like to thank the Ministry of Public Works, Infrastructures, Natural Resources and Environment (MOPIRNA) of the Democratic Republic of



São Tomé and Príncipe and the World Bank WACA programme (West Africa Coastal Area Management) for the support received during the study. In particular, we would like to thank Eng. Arlindo Carvalho, Olivio Diogo and Abnilde de Ceita Lima from the local technical team of WACA for their invaluable support, in particular during the field missions. And, finally we would like to thank Naraya Carrasco and Nicolas Desramaut from the World Bank team.

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



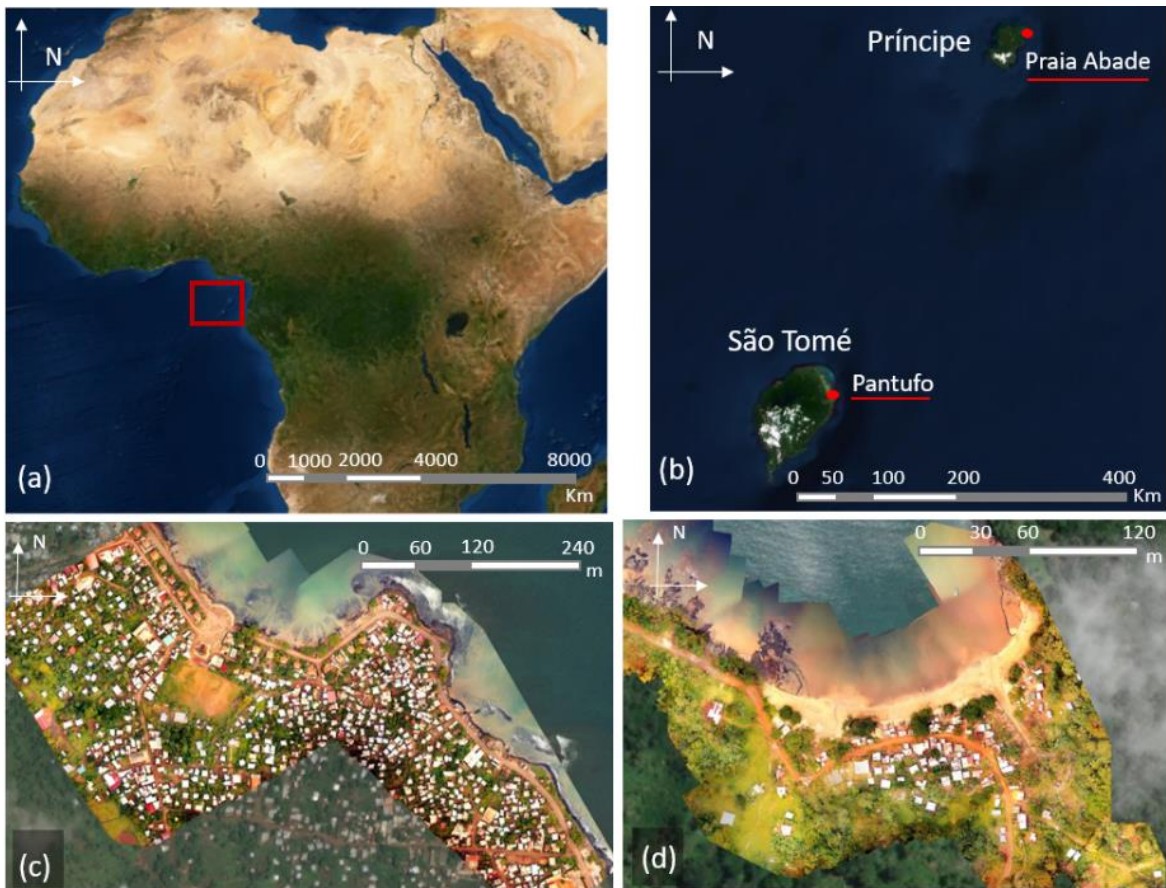

**Figure 1 Case Study site. (a) Geographical location of the islands of São Tomé and Príncipe in the Gulf of Guinea. (b) Geographical location of the communities of Pantufo and Praia Abade on the two islands. (c) Aerial view of the community of Pantufo and (d) Praia Abade. (a) and (b) are provided by ESRI, DigitalGlobe and the GIS community. (c) and (d) were reproduced with permission from CDR International.**




**Figure 2 Schematic representation of the modeling chain used to carry out the damage assessment and including the different source of uncertainties analysed (red boxes). The blue rectangles show the numerical models (XBeach and SFINCS) and tools (Delft–FIAT). The inputs used to model the hazard, exposure and vulnerability are included in orange, green and yellow boxes, respectively. Inputs used to model the present condition are represented with rectangles, whereas those used for future scenarios are represented with ellipses. Extreme sea levels are estimated by combining mean sea water levels, astronomical tides, storm surges and single waves. The earth surface is represented by bathymetric and digital elevation model (DEM) data. The inundation map produced by SFINCS is combined with depth damage functions (DDFs) and asset value to compute flood damages. Sea level rise (SLR) and socioeconomic growth are used to assess future predictions. Arrows indicate the data flow.**




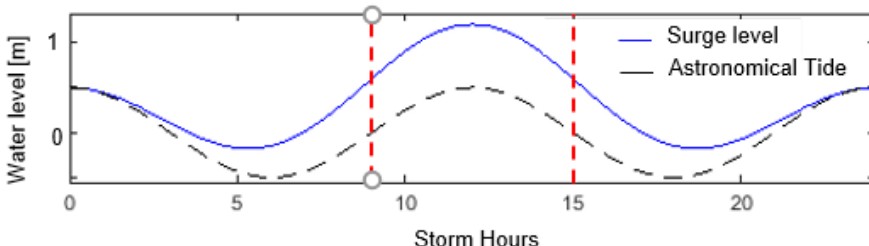

**Figure 3 Qualitative example of the hydrograph of the storm water level: surge level (blue) and astronomical tide (black) over the length of a 24 hours storm. The red lines mark the modeled central 6 hours of the storm.**

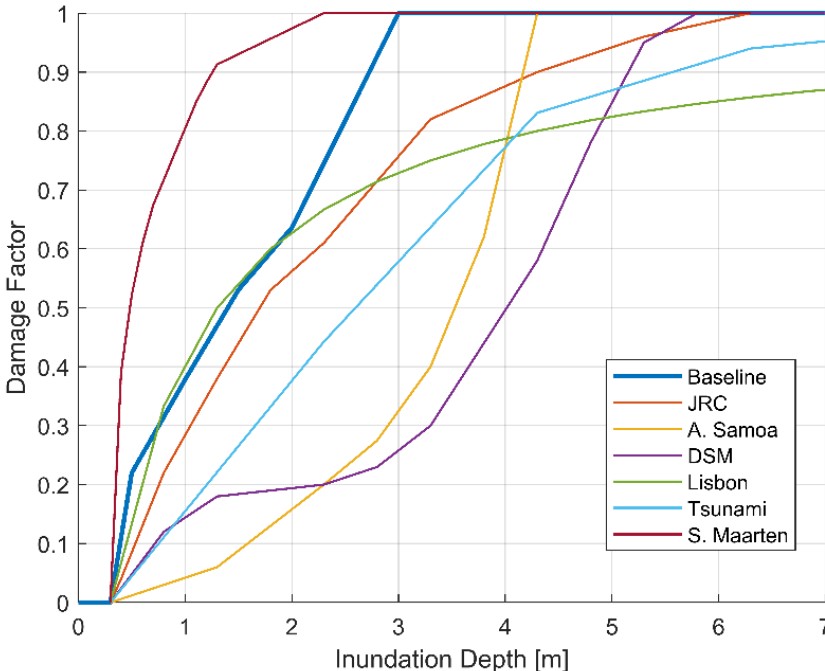

**Figure 4 Overview of the different DDFs investigated in the study, including: concave (Baseline, JRC, Lisbon, S. Maarten and Tsunami) and convex (American Samoa, and Damage Scanner Model (DSM)) types. See Table 3 for details of curves.**



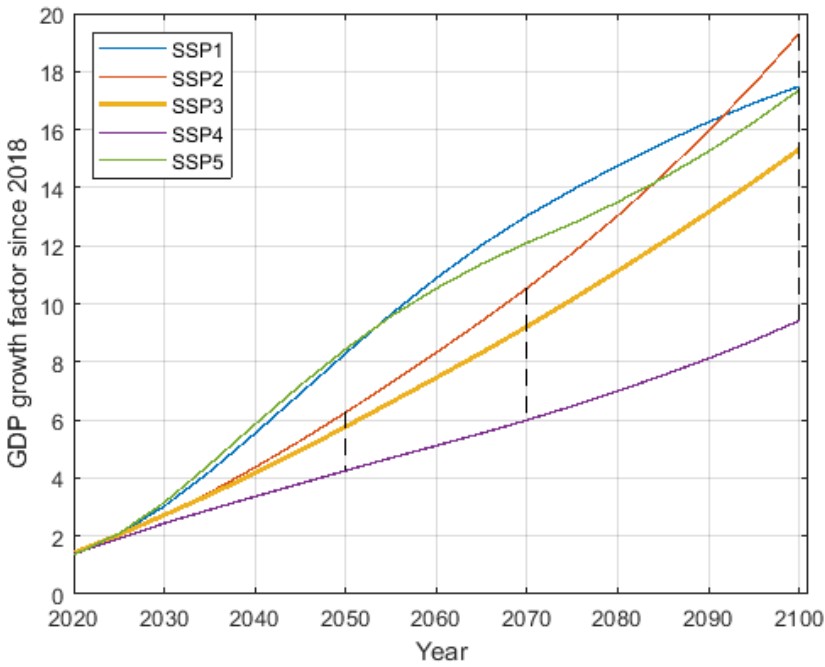

**Figure 5 GDP growth factors for five different SSP scenarios through time. The different lines indicate the projected GDP growth according to the 5 SSPs. The black dashed lines indicate the three simulated time horizons and the range of used GDP projections.**

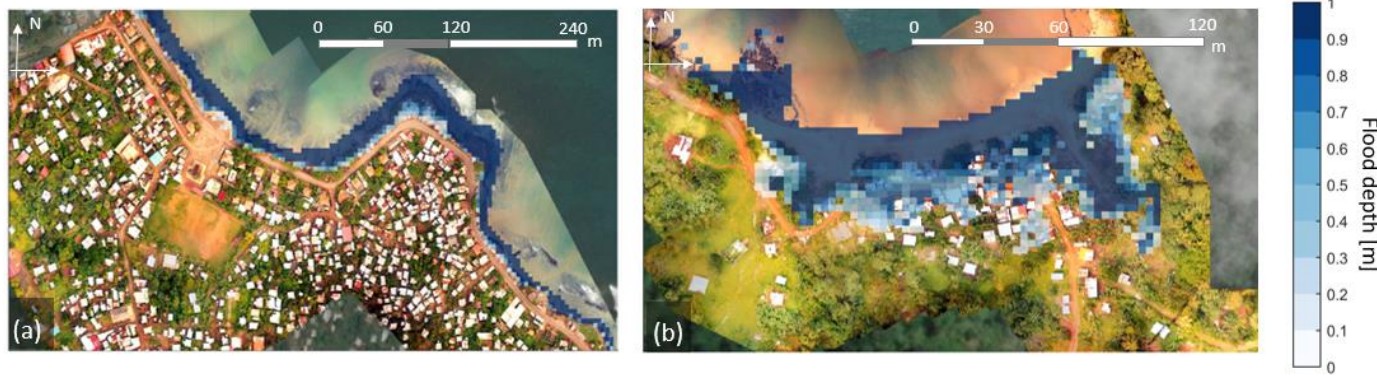

**Figure 6 Flood depth map estimated by SFINCS for the baseline scenario for Pantufo (a) and Praia Abade (b). Flood depths are expressed in meters. Both aerial images were reproduced with permission from CDR International.**


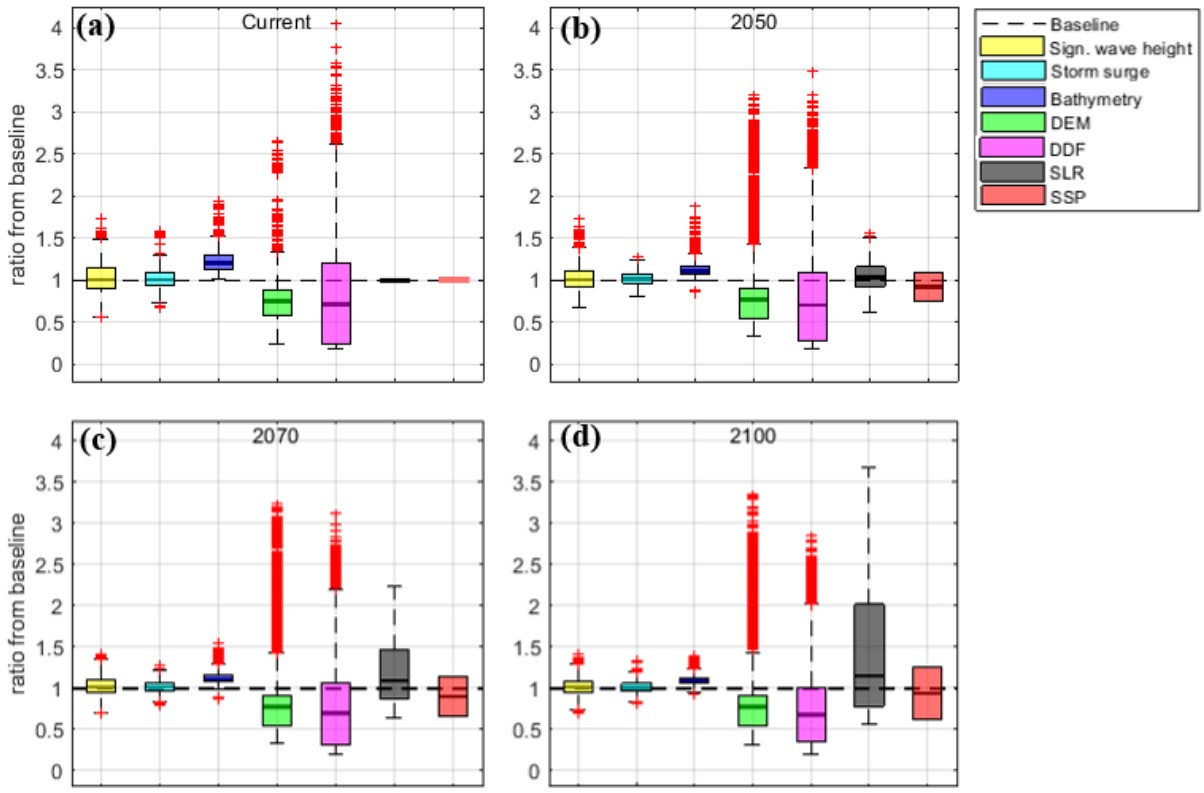

**Figure 7 Absolute impacts on damage estimate uncertainty. Box plots of the ratio of damages from the baseline scenario for Hs, storm surge, bathymetry, DEM, DDFs, SLR and SSP (a, b, c and d), for the four time horizons (present-day, 2050, 2070 and 2100). Mean values are represented by the black lines inside the boxes. The 25th and 75th percentiles are indicated by the edges of the boxes. The black thin whiskers extend to 1.5 times the interquartile distance, outside of which are outliers, shown with red crosses. The black dashed line shows the reference of the baseline scenario.**

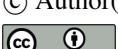
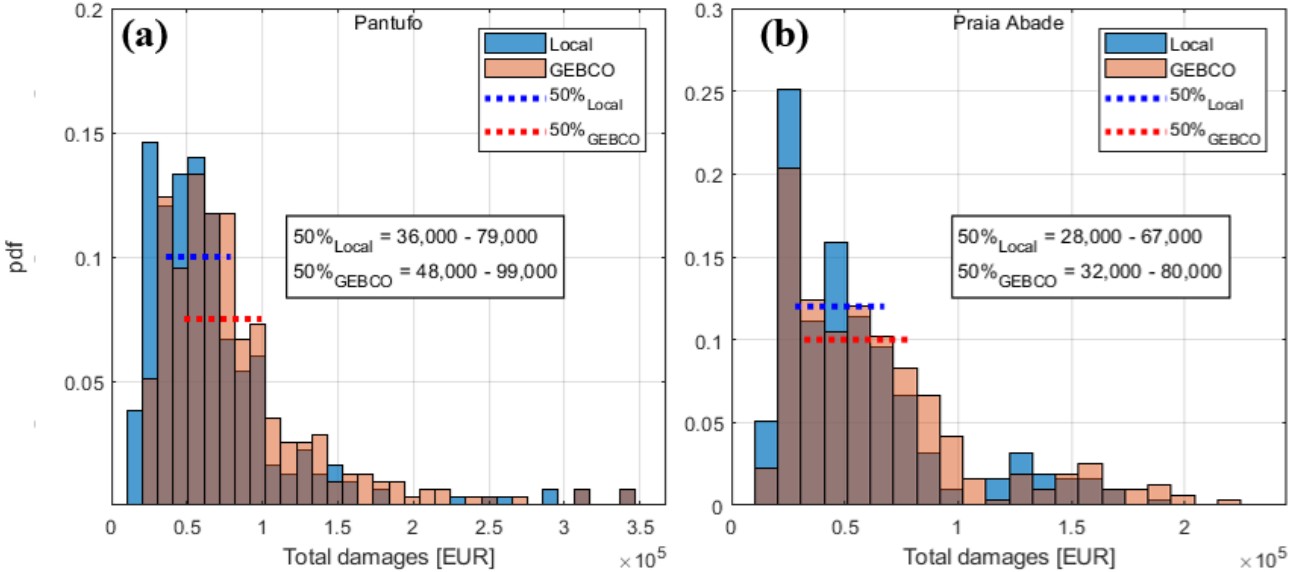

**Figure 8 Impact of using global bathymetric data versus local measured data. Histograms of damages from all 1,260 simulations of the present scenario, using a single bathymetry dataset, the locally-collected bathymetry (blue histograms) and GEBCO (orange histograms) for Pantufo (a) and Praia Abade (b). Dotted lines indicate the width of the 50 percent confidence interval. Damages are expressed in Euros (EUR).**




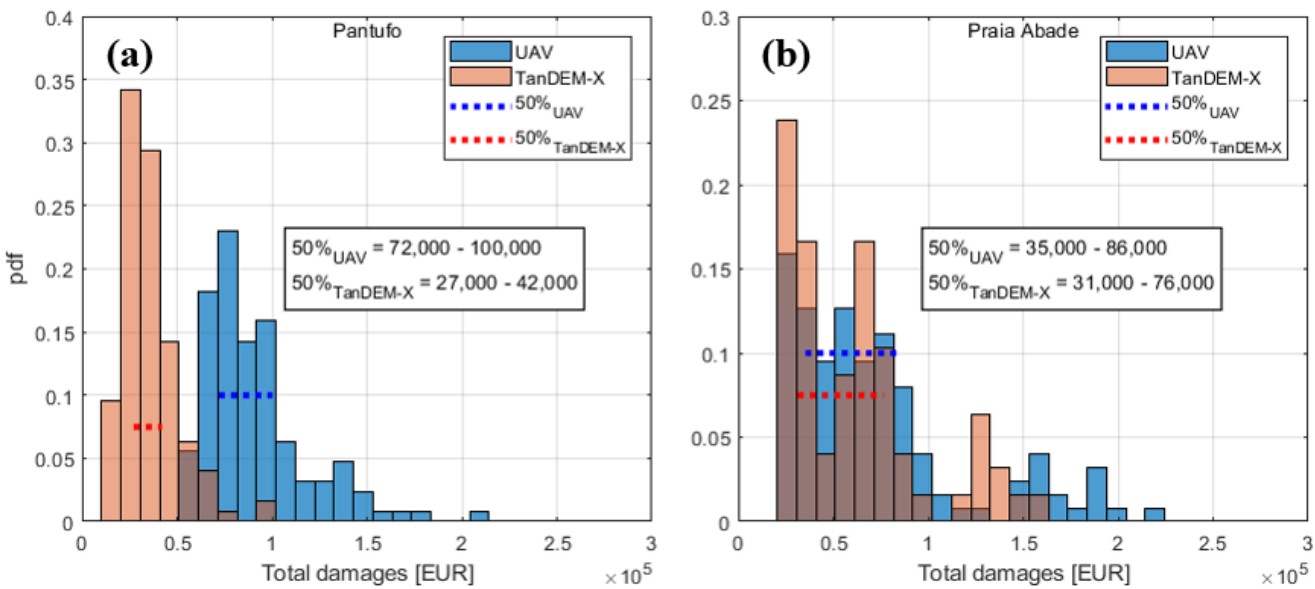

**Figure 9 Impact of using global DEMs versus local DEMs. Histograms of damages from all 1,260 simulations of the present scenario, using a single DEM dataset, the UAV-derived (blue histograms) and TanDEM-X (orange histograms) for Pantufo (a) and Praia Abade (b). Dotted lines indicate the width of the 50 percent confidence interval. Damages are expressed in Euros (EUR).**

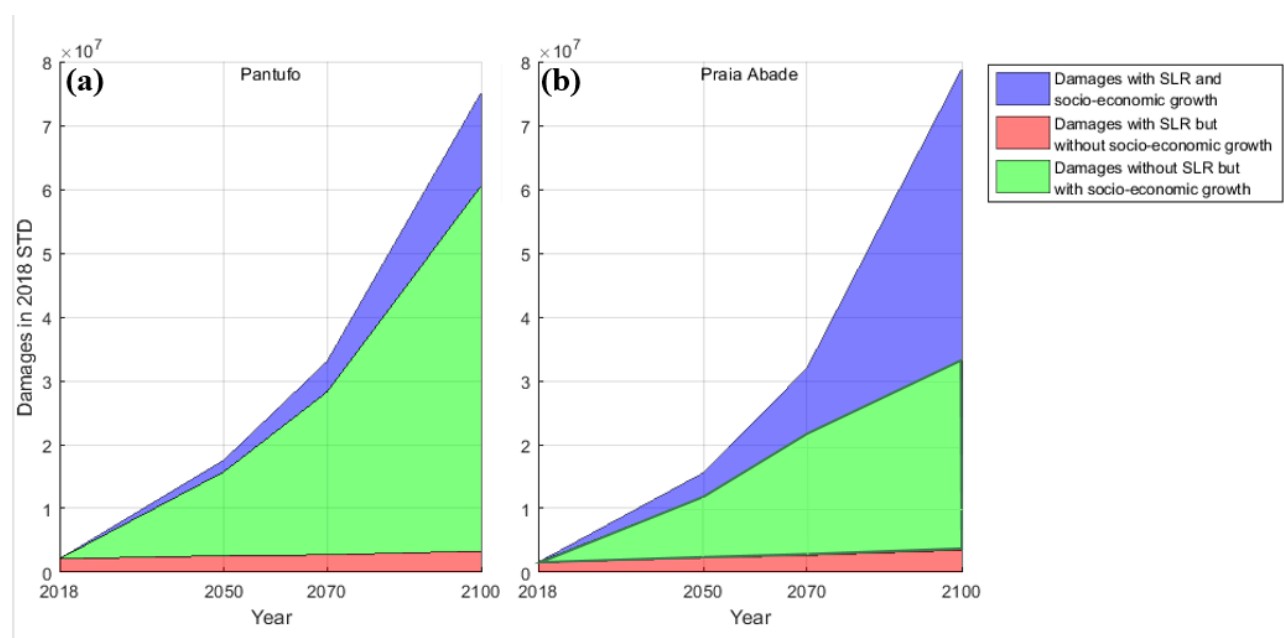

**Figure 10 The relevance of socioeconomic changes. (a) Damages under the baseline scenario expressed in Sao Tomean Dobras (STD) for Pantufo and (b) Praia Abade over time, differentiated by contributing factors: considering only climate change induced SLR (red), considering only socioeconomic changes (green) and considering both (blue).**




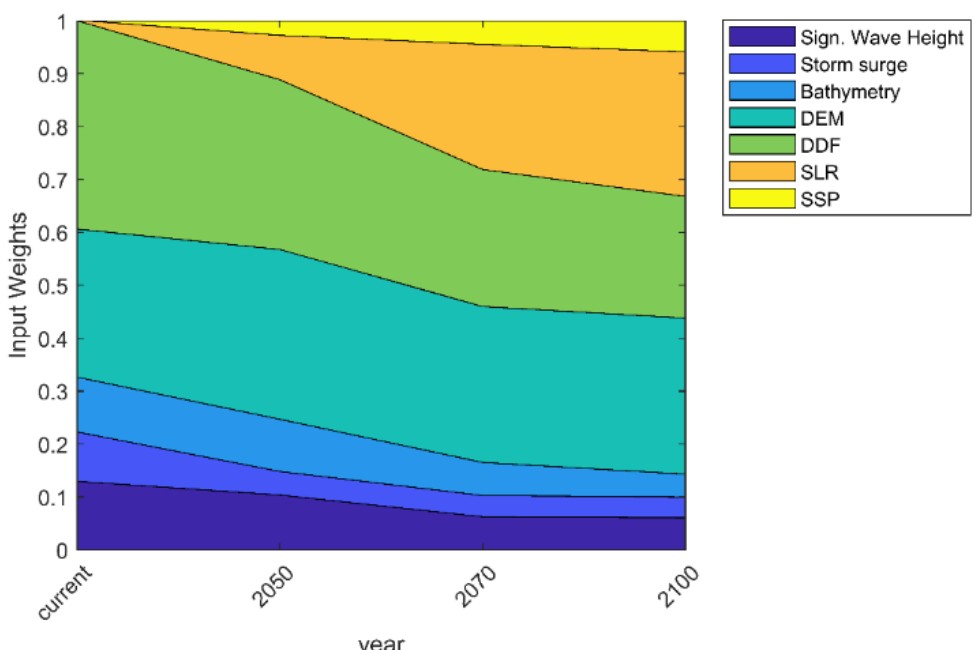

**Figure 11 Inputs relative contribution to damage estimate uncertainty. Relative weights of the investigated inputs (Hₛ, storm surge, bathymetry, DEMs, DDFs, SLR and SSP) effect on the damage estimate uncertainty over the four time horizons considered.**

5    **Table 1 Overview of all uncertainty sources investigated (input variable), with descriptions of their source of uncertainty, baseline value and the variations from the baseline value as used for the CFR analysis.**

| Input variable | Source of uncertainty | Baseline | Variations |
|---|---|---|---|
| Significant Wave Height & Storm Surge level | Uncertainty associated with the extreme value analysis (EVA) | $50^{th}$ percentile of the pdf of the extreme values (Table 2) | $5^{th}$ and $95^{th}$ percentiles of the pdf of the extreme values (Table 2) |
| Bathymetry | Horizontal and vertical resolution, errors in the dataset and interpolation between data points | Locally-measured | GEBCO |
| Digital elevation model | Horizontal and vertical resolution, errors in the dataset and interpolation between data points | Locally-measured | Multiple DEMs (The investigated satellite-derived DEMs include TanDEM-X, |



| | | | TerraSAR-X, MERIT, ASTER and SRTM. Their horizontal resolution and vertical accuracy are described in Table 3) |
|---|---|---|---|
| Depth damage function | Transfer of damage functions retrieved from other flood events and other regions. Neglect of physical factors, such as flood duration or flow velocity. | Locally-retrieved | Multiple DDFs (Table 4) |
| Sea level rise projections | Uncertainty associated with extrapolating, based on given data, as well as with reliability of climate models | Locally retrieved | Multiple DDFs (Table 4). |
| Shared Socioeconomic Pathway | Uncertainty related to future predictions of socioeconomic developments | SSP 3 – "business as usual" | SSP 2 and 4 (Fig. 5) |

**Table 2 Overview of $H_s$ and storm surge variations considered and corresponding to the 5th, 50th and 95th percentile. The baseline value is italicised.**

| Percentile | $H_s$ Praia Abade [m] | $H_s$ Pantufo [m] | Storm surge [m] |
|---|---|---|---|
| 5th | 1.05 | 1.24 | 1.05 |
| *50th* | *1.18* | *1.35* | *1.08* |
| 95th | 1.38 | 1.53 | 1.15 |

5      **Table 3 Overview of investigated globally available satellite DEMs: TanDEM-X, SRTM, MERIT, ASTER and TerraSAR-X. Horizontal resolution and global error metrics of RMSE and Mean Error (ME) for the vertical accuracy are also provided.**

| DEM | Source | Horizontal Resolution | Vertical Accuracy |
|---|---|---|---|
| | | | |





| TanDEM-X | Wessel et al. (2018) | 90 m | RMSE = 3.16 m; ME = 1.06 m (Hawker et al., 2019) |
| Shuttle Radar Topography Mission (SRTM) | Jarvis et al. (2008) | 30 m | RMSE = 4.03 m; ME = 2.16 m (Hawker et al., 2019) |
| Multi-Error Removed Improved Terrain (MERIT) | Yamazaki et al. (2017) | 90 m | RMSE = 2.32 m; ME = 1.09 m (Hawker et al., 2019) |
| Advances Spaceborne Thermal Emission and Reflection Radiometer (ASTER) | NASA/METI/AIST/Japan Spacesystems and Science (2009) | 90 m | RMSE = 8.68 m (Tachikawa et al., 2011) |
| TerraSAR-X | Produced by GeoVille in 2013, derived from TerraSAR-X imagery | 10 m | Not Available |

**Table 4 Overview of the considered depth damage functions (DDFs), their geographical application area and flood type. Different DDF curves are shown in Figure 4.**

| Depth Damage Function (DDF) | Reference | Geographical Application Area | Flood Type |
| --- | --- | --- | --- |
| JRC | Huizinga et al. (2017) | Africa | Coastal and riverine |
| S. Maarten | Vojinovic et al. (2008) | Sint Maarten (SIDS) | Coastal and pluvial |
| Lisbon | Hinkel et al. (2014) | Lisbon | Coastal |
| Tsunami | Tarbotton et al. (2015) | Averaged over several countries | Coastal (Tsunami induced) |
| Damage Scanner Model (DSM) | Kok et al. (2005) | Netherlands | Riverine |
| American Samoa | Paulik et al. (2015) | American Samoa (SIDS) | Coastal |
| Baseline | Deltares and CDR (2019) | São Tomé and Príncipe (SIDS) | Coastal and pluvial |



**Table 5 Overview of the considered SLR projections for the study area for the year 2050, 2070 and 2100, according to Vousdoukas et al. (2016). The baseline value is italicised.**

| Sea Level Rise [m] | | | |
|---|---|---|---|
| **Percentile** | **Year 2050** | **Year 2070** | **Year 2100** |
| 5th | 0.19 | 0.31 | 0.53 |
| *50th* | *0.30* | *0.49* | *0.87* |
| 95th | 0.47 | 0.98 | 2.05 |

5    **Table 6 Error metrics of the studied publicly available DEMs for the two locations. The bias and error standard deviation from the UAV-derived DEM for SRTM, MERIT, TanDEM-X, ASTER and TerraSAR-X, in Praia Abade and Pantufo.**

| Location | | SRTM | MERIT | TanDEM-X | ASTER | TerraSAR-X |
|---|---|---|---|---|---|---|
| Praia Abade | Bias [m] | 6.43 | 6.35 | 3.23 | 6.90 | Not available |
| | Error Standard Deviation [m] | 0.95 | 0.55 | 0.54 | 0.73 | Not available |
| Pantufo | Bias [m] | 4.63 | 4.48 | 2.93 | 5.81 | -1.35 |
| | Error Standard Deviation [m] | 0.97 | 0.89 | 0.68 | 0.69 | 2.30 |