# Peer review of "Uncertainties in Coastal Flood Risk Assessments in Small Island Developing States"

_Natural Hazards and Earth System Sciences, 2019_

## Referee Comment (RC1) · Anonymous Referee #1 · 19 Jan 2020

The paper assesses uncertainty in coastal flood risk from various influencing drivers in two developing island states in the Gulf of Guinea in Africa. The proposed methodology can aid in developing flood management strategies, especially in insular states. Although the study has implications for flood risk assessment, a few issues should be addressed before accepting the work, which I have summarized below: Major comments: (1) Page 4, line 20, the authors straight away analyzed a 100-year return period for the analyzed region. Although for most of the developed nations, flood protection standard assumes 100-year as the benchmark, however, for pristine locality like the two analyzed here, directly moving to a 100-year return level may lead to erroneous results based on a limited number of records available. Further, the study uses data from different sources, such as GTSR data, which is based on a 35-year record, which

may not be equivalent to the length of other available records used in the analysis. Hence, data from different sources with varying record lengths would impart additional uncertainty in the analysis. Although the return period could approximate the hazard potential beyond the record length, however, in this case, the uncertainty bound would be larger with longer return period estimates. Hence, it would be interesting to know the hazard potential on smaller return levels such as 10-year or 25-year first, that covers the analysis period before moving to the larger one.

(2) Throughout the manuscript, no-where the start year and the end year is mentioned for any of the dataset except pointing to either 30- or 35-year. In that way, it is rather abstract, what was the baseline period.

(3) For sampling extreme, authors directly have taken 98th percentile as the threshold criterion for significant wave height. For fixing threshold, authors need to ensure the sampled extremes are un-autocorrelated and iid. Further, for any coastal storms, 3-day consecutive extremes should not be sampled together. Hence, authors should ensure sampled extremes follow iid behavior.

(4) On page 4, line 15, please specify the temporal resolution of storm surges, which is 6-hourly I guess, please check. And also please explain the time frame.

(5) Please specify the source and download the link for the SSP scenario.

(6) On page 11, in the present-day scenario, interdependency between various ESL components are neglected. For example, surge and tide are often correlated (Devlin et al., 2017): Devlin, A. T., Jay, D. A., Talke, S. A., Zaron, E. D., Pan, J. and Lin, H.: Coupling of sea level and tidal range changes, with implications for future water levels, Scientific reports, 7(1), 17021, 2017.

(7) In page 11, please also include the factor, "technological advancement" since this also affects future flood risk assessment.

Minor comments: (1) Authors have cited several studies related to coastal and riverine

flood risk assessment and associated uncertainties. However, in the low-lying deltas, the compound flooding resulting from both coastal and riverine floods during a major storm episode can cause significant damage. A few references on this could broaden the scope of the article.

(2) Page 4, line 10, no citation is available for De Ridder et al., 2019 in the reference section.

---

## Referee Comment (RC2) · Anonymous Referee #2 · 5 Feb 2020

The paper is interesting and definitely of interest to the journal's audience. Overall I am positive but I think some work is needed to make it publication-ready. The main issue is that the manuscript gives the impression to have been written in a hurry, as several aspects of the methodology and the findings are not being properly explained. More specifically: The methodology to estimate the uncertainty is not clearly described. Do the authors simulate all possible combinations? How do they discretize their parameter space? I assume that they keep all factors but the tested one to 'baseline' values and allow only the tested parameter to fluctuate. However, this is not clearly explained in the manuscript. Also in what way does the bathymetry and Hs result in damages? Section 3.2.2: Considering only SLR is an acceptable assumption but the authors should at least mention other studies which discuss the other uncertainty sources (astronomical

and meteorological tides). Note also that the text is almost the same as in the discussion (page 12, line 7). SSPs: there is literature on the compatibility of certain SSPs and RCPs and the authors should justify why they combine SSP2 with RCP8.5. The use of 5th and 95th percentile values is an assumption which may be inevitable but has limitations and should be discussed. It's not said that the 95th Hs will result in the 95th damage and to assess this properly a Monte Carlo framework is needed considering the whole PDF and more cases. I expect that the computational effort is prohibitive but this should be at least discussed. Also as the methodology is not sufficiently explained it is not easy to follow what simulations have been really done. The Hs uncertainty sources are not fully covered. Given that values come from a reanalysis one should also include model error and EVA uncertainties beyond the one related to fitting (e.g. other pdfs). These aspects should be discussed. All bathymetry related uncertainty is also not addressed: e.g. effect on wave/storm surge simulations. Again discuss

Other minor comments: I would recommend expressing EAD in USD since EUR is not relevant to people living outside of the EU and in this case so the study areas are also out of EU territory. Page 1, Line 24: Rephrase 'which challenges the safety and sustainability of their society and the growth of their economies' page 2, Line 18: 2017)and CoastalDEM page 4, Line 20: I am a bit skeptical about whether direct damages are dominant. I would suggest removing this statement unless the authors can support it with references/data Page 6, Line 1: The work of Bove et al (https://doi.org/10.1016/j.scitotenv.2019.136162) is also relevant to the present study and should be discussed. Page 6, Line 26: correct 'being most representatives' Table 1: SLR projections columns 3 and 4, is this correct? Also it is not clear where exactly SLR comes from. Vousdoukas et al. 2016 is cited but not with sufficient details. Maybe the data come from the 2018 Nature Communications paper? Figure 7 would be easier to follow with x labels explaining each bar Explain the vertical datum used for the UAV DEM Figure 10 compares the damages driven by socioeconomic vs climate change so I would suggest expressing that way

---

## Author Comment (AC1) · 1 Apr 2020

Dear Editor and Dear Reviewers,

We appreciate the reviewers' constructive comments. We addressed the comments and suggestions in this reply and have accordingly implemented changes in the manuscript, which have surely improved its quality. Please do not hesitate to contact me if you have further questions.

Kind Regards,

Matteo Parodi

Reviewer #1

1) Page4, line20, the authors straight away analysed a 100-year return period for the analyzed region. Although for most of the developed nations, flood protection standard assumes 100-year as the benchmark, however, for pristine locality like the two analyzed here, directly moving to a 100-year return level may lead to erroneous results based on a limited number of records available. Further, the study uses data from different sources, such as GTSR data, which is based on a 35-year record, which may not be equivalent to the length of other available records used in the analysis. Hence, data from different sources with varying record lengths would impart additional uncertainty in the analysis. Although the return period could approximate the hazard potential beyond the record length, however, in this case, the uncertainty bound would be larger with longer return period estimates. Hence, it would be interesting to know the hazard potential on smaller return levels such as 10-year or 25-year first that covers the analysis period before moving to the larger one.

Authors: We thank the reviewer for the very good comment. We analysed different return periods and found that, for the considered locations, smaller return periods had a very little difference in the range of values between the 5th and 95th percentile values (13 cm spread for the 10-year return value and 28 cm spread for the 100-year return value for offshore significant wave height; 5 cm spread for the 10-year return value and 8 cm spread for the 100-year return value for storm surge level). The estimated values for different return periods for the two variables (i.e. significant wave height and storm surge levels) are shown in Fig. 1 and Fig. 2 which have been added for completeness to the rebuttal letter. In the top left of the figure, the 50th percentile, together with the 5th and 95th percentile values in brackets, are shown for each return period.

Furthermore, the 50th percentile value did not increase significantly from a 1-year return value to a 100-year return value for both investigated variables (1.10 m to 1.35 m for offshore significant wave height; 1 m to 1.08 m for storm surge level). Therefore, the hazard potential on smaller return periods (e.g. 10-year or 25-year) would have not differed significantly from the one computed using the 100-year return period. Moreover, the 5th to 50th percentile values range for the 100-year return period, which was analysed as part of this paper, included the 50th percentile value for the 10-year return period. The choice of the 100-year return period, instead of the use of other return periods, was arbitrary and with the only objective to investigate uncertainty. This has been clarified in the manuscript: "In this analysis, arbitrarily chosen 100-year return period extreme sea levels event were modelled since, for this case, events with smaller return periods had only a small difference in intensity and computed flood damages than the 100-year return period event. ", see page 4, line 25.

We agree that using data with varying record lengths would impart additional uncertainty on the analysis and the issue has been mentioned: "Separate datasets with different recorded lengths were used for the statistical estimation of the storm surge level and significant wave height 100-year return period values, which is an additional source of uncertainty in the damage prediction", see page 12, line 16.

(2) Throughout the manuscript, no-where the start year and the end year is mentioned for any of the dataset except pointing to either 30- or 35-year. In that way, it is rather abstract, what was the baseline period.

Authors: The issue has been addressed and the start and end dates of each dataset that we used are mentioned on page 5 line 7 and 22 for significant wave height and storm surge, respectively. "The ERA-Interim dataset (Dee et al. 2011) by ECMWF (European Centre for Medium-Range Weather Forecast), which covers the period from January 1st, 1989 until present, was used."

"The estimation of storm surge levels was based on the dataset by Muis et al. (2016), a global water level reanalysis based on daily maxima over the time period 1979-2014."

(3) For sampling extreme, authors directly have taken 98th percentile as the threshold criterion for significant wave height. For fixing threshold, authors need to ensure the sampled extremes are un-autocorrelated and iid. Further, for any coastal storms,3-day consecutive extremes should not be sampled together. Hence, authors should

ensure sampled extremes follow iid behavior.

Authors: The choice of the threshold was based on literature (Wahl et al. 2017), and the 98th percentile was chosen as the authors recommended. Furthermore, coastal storms have been sampled with a 60-hours span time in between, in order to, as rightly mentioned by the reviewer, ensure that sampled extremes followed iid (independent and identically distributed) behaviour and were not autocorrelated. This has been specified in the manuscript: "To ensure the clustered peaks were independent and identically distributed, 60 hour consecutive extremes were not sampled together.", see page 5, line 11.

(4) On page 4, line 15, please specify the temporal resolution of storm surges, which is 6-hourly I guess, please check. And also please explain the time frame.

Authors: The time frame of the investigated storm surge are the 6 hours peak of a 24hour storm. The temporal resolution is one hour. This information has been specified accordingly on page 4, line 9.

(5) Please specify the source and download the link for the SSP scenario.

Authors: The source has been specified on page 8, line 2 and a footnote has been made for the download link: https://tntcat.iiasa.ac.at/SspDb/dsd?Action=htmlpage&page=about.

(6) On page 11, in the present-day scenario, interdependency between various ESL components are neglected. For example, surge and tide are often correlated (Devlin et al., 2017): Devlin, A. T., Jay, D. A., Talke, S. A., Zaron, E. D., Pan, J. and Lin, H.: Coupling of sea level and tidal range changes, with implications for future water levels, Scientïfic reports, 7(1), 17021, 2017.

Authors: Indeed, in the presented work ESL components are considered independent. The authors have assumed that storm surge and tidal level, as well as offshore wave height and tidal level are independent. However, as the reviewer noted, interdependency between various ESL components can occur. The issue has been mentioned as a limitation of the study: "Moreover, the interdependency between different ESL components has been neglected, although tide and sea level changes are often correlated, adding further uncertainty in the analysis (Devlin et al. 2017)" In Section 5, page 12, line 12.

(7) In page 11, please also include the factor, "technological advancement" since this also affects future flood risk assessment.

Authors: We agree that technological advancement may also affect future flood risk assessment and it has been mentioned on page 12, line 23.

Minor comments:

(1) Authors have cited several studies related to coastal and riverine flood risk assessment and associated uncertainties. However, in the low-lying deltas, the compound flooding resulting from both coastal and riverine floods during a major storm episode can cause significant damage. A few references on this could broaden the scope of the article.

Authors: We agree with the comment. Indeed, compound flooding can increase significantly the damages than if they had occurred for a coastal or riverine flood alone. The topic has been mentioned: "Compound flooding events (e.g. coastal and riverine) can significantly increase the damages than single events only (Kumbier et al., 2018; Wahl et al., 2015; Ward et al., 2017), and further research could estimate the added uncertainty.", see Section 5, page 12, line 10.

(2) Page 4, line 10, no citation is available for De Ridder et al., 2019 in the reference section

Authors: The reference has been removed, since the cited manuscript is still under review.

Please also note the supplement to this comment:
https://www.nat-hazards-earth-syst-sci-discuss.net/nhess-2019-392/nhess-2019-392-AC1-supplement.pdf

—————————————————————

[Figure]

Return values

1 year: 110.57 ( 108.59, 112.52 )
2 year: 115.09 ( 112.24, 117.83 )
5 year: 120.58 ( 116.10, 125.18 )
10 year: 124.41 ( 118.61, 131.12 )
50 year: 132.31 ( 123.09, 145.66 )
100 year: 135.33 ( 124.51, 152.21 )

**Fig. 1.** Figure 1 Estimated Significant wave height values for different return periods. Values are expressed in cm. The black crosses show the 50th percentile. The black dotted lines show the lower and upper

Return values
1 year: 100.80 ( 99.59, 102.06 )
10 year: 106.48 ( 104.19, 108.88 )
50 year: 108.15 ( 104.81, 112.59 )
100 year: 108.58 ( 104.91, 114.10 )

Return period [years]

**Fig. 2.** Figure 2 Estimated Storm surge level values for different return periods. Values are expressed in cm. The black crosses show the 50th percentile. The black dotted lines show the lower and upper bounds

**Supplement:**

**Reply letter manuscript "Uncertainties in Coastal Flood Risk Assessments in Small Island Developing States"**

Dear Editor and Dear Reviewers,

We appreciate the reviewers' constructive comments. We addressed the comments and suggestions in this reply and have accordingly implemented changes in the manuscript, which have surely improved its quality. Please do not hesitate to contact me if you have further questions.

Kind Regards,

Matteo Parodi

Reviewer #1

1) Page4, line20, the authors straight away analysed a 100-year return period for the analyzed region. Although for most of the developed nations, flood protection standard assumes 100-year as the benchmark, however, for pristine locality like the two analyzed here, directly moving to a 100-year return level may lead to erroneous results based on a limited number of records available.

Further, the study uses data from different sources, such as GTSR data, which is based on a 35-year record, which may not be equivalent to the length of other available records used in the analysis. Hence, data from different sources with varying record lengths would impart additional uncertainty in the analysis. Although the return period could approximate the hazard potential beyond the record length, however, in this case, the uncertainty bound would be larger with longer return period estimates. Hence, it would be interesting to know the hazard potential on smaller return levels such as 10-year or 25-year first, that covers the analysis period before moving to the larger one.

We thank the reviewer for the very good comment. We analysed different return periods and found that, for the considered locations, smaller return periods had a very little difference in the range of values between the $5^{th}$ and $95^{th}$ percentile values (13 cm spread for the 10-year return value and 28 cm spread for the 100-year return value for offshore significant wave height; 5 cm spread for the 10-year return value and 8 cm spread for the 100-year return value for storm surge level). The estimated values for different return periods for the two variables (i.e. significant wave height and storm surge levels) are shown in Fig. 1 and Fig. 2 which have been added for completeness to the rebuttal letter. In the top left of the figure, the $50^{th}$ percentile, together with the $5^{th}$ and $95^{th}$ percentile values in brackets, are shown for each return period.

Furthermore, the $50^{th}$ percentile value did not increase significantly from a 1-year return value to a 100-year return value for both investigated variables (1.10 m to 1.35 m for offshore significant wave height; 1 m to 1.08 m for storm surge level). Therefore, the hazard potential on smaller return periods (e.g. 10-year or 25-year) would have not differed significantly from the one computed using the 100-year return period. Moreover, the $5^{th}$ to $50^{th}$ percentile values range for the 100-year return period, which was analysed as part of this paper, included the $50^{th}$ percentile value for the 10-year return period.
The choice of the 100-year return period, instead of the use of other return periods, was arbitrary and with the only objective to investigate uncertainty. This has been clarified in the manuscript: "In this analysis, arbitrarily chosen 100-year return period extreme sea levels event were modeled since, for this case, events with smaller return periods had only a small difference in intensity and computed flood damages than the 100-year return period event. ", see page 4, line 25.

We agree that using data with varying record lengths would impart additional uncertainty on the analysis and the issue has been mentioned: "Separate datasets with different recorded lengths were used for the

statistical estimation of the storm surge level and significant wave height 100-year return period values, which is an additional source of uncertainty in the damage prediction", see page 12, line 16.

[Figure]

Figure 1 Estimated Significant wave height values for different return periods. Values are expressed in cm. The black crosses show the 50th percentile. The black dotted lines show the lower and upper bounds of the 90% confidence interval (the 5th and 95th percentile values). In the top left corner, the 50th and within brackets, the 5th and 95th values, are reported for the 1, 10, 50 and 100-year return period.

[Figure]

Figure 2 Estimated Storm surge level values for different return periods. Values are expressed in cm. The black crosses show the 50th percentile. The black dotted lines show the lower and upper bounds of the 90% confidence interval (the 5th and 95th percentile values). In the top left corner, the 50th and, within brackets, the5th and 95th values, are reported for the 1, 10, 50 and 100-year return period.

(2) Throughout the manuscript, no-where the start year and the end year is mentioned for any of the dataset except pointing to either 30- or 35-year. In that way, it is rather abstract, what was the baseline period.

The issue has been addressed and the start and end dates of each dataset that we used are mentioned on page 5 line 7 and 22 for significant wave height and storm surge, respectively. "The ERA-Interim dataset (Dee et al. 2011) by ECMWF (European Centre for Medium-Range Weather Forecast), which covers the period from January 1st, 1989 until present, was used."

"The estimation of storm surge levels was based on the dataset by Muis et al. (2016), a global water level reanalysis based on daily maxima over the time period 1979-2014."

(3) For sampling extreme, authors directly have taken 98th percentile as the threshold criterion for significant wave height. For fixing threshold, authors need to ensure the sampled extremes are unautocorrelated and iid. Further, for any coastal storms,3-day consecutive extremes should not be sampled together. Hence, authors should ensure sampled extremes follow iid behavior.

The choice of the threshold was based on literature (Wahl et al. 2017), and the 98th percentile was chosen as the authors recommended. Furthermore, coastal storms have been sampled with a 60-hours span time in between, in order to, as rightly mentioned by the reviewer, ensure that sampled extremes followed iid (independent and identically distributed) behaviour and were not autocorrelated. This has been specified in the manuscript: "To ensure the clustered peaks were independent and identically distributed, 60 hour consecutive extremes were not sampled together.", see page 5, line 11.

(4) On page 4, line 15, please specify the temporal resolution of storm surges, which is 6-hourly I guess, please check. And also please explain the time frame.

The time frame of the investigated storm surge are the 6 hours peak of a 24hour storm. The temporal resolution is one hour. This information has been specified accordingly on page 4, line 9.

(5) Please specify the source and download the link for the SSP scenario.

The source has been specified on page 8, line 2 and a footnote has been made for the download link: https://tntcat.iiasa.ac.at/SspDb/dsd?Action=htmlpage&page=about.

(6) On page 11, in the present-day scenario, interdependency between various ESL components are neglected. For example, surge and tide are often correlated (Devlin et al., 2017): Devlin, A. T., Jay, D. A., Talke, S. A., Zaron, E. D., Pan, J. and Lin, H.: Coupling of sea level and tidal range changes, with implications for future water levels, Scientific reports, 7(1), 17021, 2017.

Indeed, in the presented work ESL components are considered independent. The authors have assumed that storm surge and tidal level, as well as offshore wave height and tidal level are independent. However, as the reviewer noted, interdependency between various ESL components can occur. The issue has been mentioned as a limitation of the study: "Moreover, the interdependency between different ESL components has been neglected, although tide and sea level changes are often correlated, adding further uncertainty in the analysis (Devlin et al. 2017)" In Section 5, page 12, line 12.

(7) In page 11, please also include the factor, "technological advancement" since this also affects future flood risk assessment.

We agree that technological advancement may also affect future flood risk assessment and it has been mentioned on page 12, line 23.

**Minor comments:**

(1) Authors have cited several studies related to coastal and riverine flood risk assessment and associated uncertainties. However, in the low-lying deltas, the compound flooding resulting from both coastal and riverine floods during a major storm episode can cause significant damage. A few references on this could broaden the scope of the article.

We agree with the comment. Indeed, compound flooding can increase significantly the damages than if they had occurred for a coastal or riverine flood alone. The topic has been mentioned: "Compound flooding events (e.g. coastal and riverine) can significantly increase the damages than single events only (Kumbier et al., 2018; Wahl et al., 2015; Ward et al., 2017), and further research could estimate the added uncertainty.", see Section 5, page 12, line 10.

(2) Page 4, line 10, no citation is available for De Ridder et al., 2019 in the reference section

The reference has been removed, since the cited manuscript is still under review.

Reviewer #2

Overall I am positive but I think some work is needed to make it publication-ready. The main issue is that the manuscript gives the impression to have been written in a hurry, as several aspects of the methodology and the findings are not being properly explained.

More specifically: The methodology to estimate the uncertainty is not clearly described. Do the authors simulate all possible combinations?

We appreciate the reviewers' comment and apologize that the methodology and findings do not seem properly explained. To address this comment, we have tried to add more detailed explanation on the methodology and data used.
With regard to the specific comment we added on page 8, line 14 that: "We conducted a sensitivity analysis on the full parameter space of model inputs. This led to combinations of: (a) 3 $H_s$ scenarios (b) 3 storm surge scenarios (c) 2 bathymetry scenarios (d) 6 DEM scenarios (e) 7 DDF scenarios (f) 3 SLR scenarios (g) 3 SSP scenarios over (h) 4 different time horizons (current[1], 2050, 2070 and 2100) ultimately leading to a total of 21,168 simulations for each community." The number of variations (scenarios) for each different variable has also been added to Table 1.

How do they discretize their parameter space? I assume that they keep all factors but the tested one to 'baseline' values and allow only the tested parameter to fluctuate. However, this is not clearly explained in the manuscript.

The parameter space has been discretized differently for each input. The different values used for each input are described in Table 1. Some inputs were represented by a probability distribution (e.g. storm surge and significant wave height). However, some other inputs (e.g. DEM and Bathymetry) were not represented by a probability distribution but we used the different datasets that were publicly available. In this case, for each input, the parameter space is composed by the selected publicly available datasets. The model simulations were run testing all parameter's combinations, as explained in the above comment, and not with a one at a time (OAT) method where all parameters but the tested one are kept at the baseline value.

Also in what way does the bathymetry and Hs result in damages?

Varying the bathymetry and Hs input values affects the computed flooding damages. Increasing Hs leads to an increase in wave-induced flooding and thus increased damages. Higher bathymetry values will reduce flooding and hence computed damages. The changes in the computed damages described in Section 4 and depicted in Figure 7 illustrate this situation.

Section 3.2.2: Considering only SLR is an acceptable assumption but the authors should at least mention other studies which discuss the other uncertainty sources (astronomical and meteorological tides). Note also that the text is almost the same as in the discussion (page 12, line 7).

We agree with the comment and included in the discussion studies that have considered the other uncertainty sources mentioned (Chowdhury et al., 2006; Karim and Nobuo, 2008), see page 7, line 15. The text in Section 5 expresses the same concept as in Section 3.2.2. In Section 3.2.2, the assumption is first introduced, while in Section 5 is expressed again in the discussion in order to further underline the importance of its implications of this assumption.
* * *
[1] For the current time horizon, no SLR and SSP scenarios are present, reducing the number of simulations required

SSPs: there is literature on the compatibility of certain SSPs and RCPs and the authors should justify why they combine SSP2 with RCP8.5.

We acknowledge that different SSPs are compatible with different RCPs, according to literature. However, RCP are global scenarios and individual countries, especially smaller ones, could have a combination of SSP and RCP which would not be logical or possible at a local scale. We have clarified this in the manuscript: "Although some SSP scenarios are only compatible with certain RCP scenarios at the global or regional scale (van Vuuren and Carter, 2014), at the local scale of individual and small countries RCP and SSP may not be necessarily correlated, since RCP represent a global process while SSP reflect the socioeconomic development of the single country. ", see page 8, line 10.

The use of 5$^{th}$ and 95$^{th}$ percentile values is an assumption which may be inevitable but has limitations and should be discussed. It is not said that the 95th Hs will result in the 95th damage and to assess this properly a Monte Carlo framework is needed considering the whole PDF and more cases. I expect that the computational effort is prohibitive but this should be at least discussed.

The issues have been mentioned and added "To further improve the presented methodology, a Monte Carlo analysis that considers a pdf for each uncertain input to estimate the pdf of the expected damages could be performed, although the computational effort is prohibitive", in Section 5, page 12, line 2.

Also as the methodology is not sufficiently explained it is not easy to follow what simulations have been really done.

This comment is similar to Reviewer 2's first one, and we have addressed it there and on page 8, line 24.

The Hs uncertainty sources are not fully covered. Given that values come from a reanalysis one should also include model error and EVA uncertainties beyond the one related to fitting (e.g. other pdfs). These aspects should be discussed.

We have included EVA uncertainties, related to estimating values for a return period longer than the length of the available data. "Commonly, extreme hydrodynamic boundary conditions are represented with probability distributions. However, these distributions are fit to measured data and attempt to estimate values for return periods longer than the length of the available data, thus already introducing uncertainty in the model.", see page 5, line 13.

Our analysis focused on uncertainty related to input values and model uncertainty and error were not taken into account. However, we have mentioned the Hs uncertainty brought by model error in the analysis. "Furthermore, the nearshore wave conditions were estimated from transformation matrices in the DELFT3D-WAVE (SWAN) model, which increases the uncertainty of Hs by introducing model errors." See page 5 line 15.

All bathymetry related uncertainty is also not addressed: e.g. effect on wave/storm surge simulations. Again discuss

The discussion on bathymetry related uncertainties has been improved. "Therefore, uncertainty and errors in bathymetric datasets could lead to an increased uncertainty in wave and storm surge simulations, increasing the potential for modeling error or biases.", see page 5, line 28.

Other minor comments:

I would recommend expressing EAD in USD since EUR is not relevant to people living outside of the EU and in this case so the study areas are also out of EU territory.

The objective of the manuscript is to show the relative importance of the considered input uncertainties on the estimated flood damages and the relative changes in damages, so we have decided not to change the currency used

Page 1, Line 24: Rephrase 'which challenges the safety and sustainability of their society and the growth of their economies'

This has been rephrased on page 1 line 27.

page 2, Line 18: 2017) and CoastalDEM

This has been corrected on page 2, line 25.

page 4, Line 20: I am a bit skeptical about whether direct damages are dominant. I would suggest removing this statement unless the authors can support it with references/data

The statement has been removed.

Page 6, Line 1: The work of Bove et al (https://doi.org/10.1016/j.scitotenv.2019.136162) is also relevant to the present study and should be discussed.

Thank you, we now cite this paper on page 6, line 13.

Page 6, Line 26: correct 'being most representatives'

The sentence has been corrected on page 7, line 7.

Table 1: SLR projections columns 3 and 4, is this correct? Also it is not clear where exactly SLR comes from. Vousdoukas et al. 2016 is cited but not with sufficient details. Maybe the data come from the 2018 Nature Communications paper?

The Table columns have been corrected. Also, the citation of the data has been updated to provide enough details to access the sea level rise predictions data on page 7, line 21.

Figure 7 would be easier to follow with x labels explaining each bar

The Figure has been updated accordingly, including x labels.

Explain the vertical datum used for the UAV DEM

The UAV DEM was referenced to the WGS84 datum, and we have updated the paper including this information on page 6, line 18.

Figure 10 compares the damages driven by socioeconomic vs climate change so I would suggest expressing that way

The Figure caption has been updated accordingly.

---

## Author Response (AR2)

**Reply letter manuscript "Uncertainties in Coastal Flood Risk Assessments in Small Island Developing States"**

Dear Editor and Dear Reviewers,

We appreciate the reviewers' second round of comments. In this reply, we have addressed all the comments and suggestions. The changes have been implemented in track changes on the original manuscript, improving its quality and clarity. Please do not hesitate to contact me if you have further questions.

Kind Regards,

Matteo Parodi

Reviewer #1

The paper incorporates most of my earlier comments. However, still, a few things need to be clarified. The paper analyses uncertainty among different ESL components and found DEM was the main uncertain component. However, no significant statistical tests were performed to partition sources of uncertainty.

This comment is addressed below at point 8 and 9.

1. In Abstract, line 14 "input data with uncertain accuracy" replace with "input uncertainty". Further, in Abstract, line 23, "predictive error"; this is not an error, it is "predictive uncertainty"

We have updated the Abstract accordingly to this comment, at line 14 and 23.

2. In page 2, Line 26, double period.

The second period has been removed.

3. In page 2, Line 31, "…often damage curves are taken from literature and applied in different areas making few if any, adjustments…" In your case as well, you have taken return level estimates from Muis et al. which is again from disparate data sources and directly implemented for your analyses. So what are the difference b/n second order (where damage curve being deployed from different sources) and first order (in your case, you are taking the specific quantile corresponds to 100-yr RL) sources of uncertainty?

The focus of the paper is looking into uncertainties of different input data sources rather than on assumption within the models. This is mentioned in the discussion: "However, each model contains numerous assumptions and simplification that translate into further uncertainties in the output estimate (Loucks and Van Beek, 2017; Uusitalo et al., 2015). These model uncertainties were disregarded as we focused only on uncertainties related to data input." Page 12, line 2.

4. In page 3, line 20, any literature that points to the vulnerability of coastal flooding in the area considered? Please cite

A citation to the geomorphology study (Deltares and CDR, 2019), already cited in the manuscript, has been made in page 3, line 21. Furthermore, another previous study that points to the vulnerability to coastal flooding of São Tomé has been cited on page 3, line 23 (Giardino et al., 2012).

5. Page 4, line 27, the small differences in intensity at different return levels could be shown in the appendix

The different return level values are shown in the two footnotes on Page 5, referenced at line 15 and 29.

"[1] 10-year period 50th percentile value and 90 percent confidence interval values: 1.24 m (1.19-1.31 m)
[1] 10-year period 50th percentile value and 90 percent confidence interval values: 1.06 m (1.04-1.08 m)"

6. in Page 5, line 7, for significant wave-height, please provide the information regarding variables used in parentheses; authors possibly used "significant height of combined wind waves and swell" at 6-hourly timescale archived at ERA-Interim. But it would be wise to mention that for further reproducibility of the result.

The information regarding variables used for significant wave-height has been specified in the manuscript: "The dataset provides 6-hourly significant wave height ($H_s$) of combined wind and swell data", page 5, line 14.

7. In page 7, Line 21, the statement "The choice of RCP 8.5 relies on the fact that the 90% confidence interval of this projection also captures most of the 90% confidence interval values under the RCP 4.5 scenario." This statement is not so clear, RCP 8.5 is altogether a different scenario when GHG emission reaches 8.5 W7m2 and considered to be most extreme; an overlap b/n two scenario may misrepresent risk.

For the location considered, the 90% confidence interval values of SLR projections under the RCP 8.5 scenario captures most of the 90% confidence interval of SLR projections under the RCP 4.5 scenario. This has been better explained in the text: "The choice of RCP 8.5 relies on the fact that the 90% confidence interval of SLR projections under this scenario also captures most of the 90% confidence interval of SLR projections under the RCP 4.5 scenario.", see page 7, line 30.

8. In page 11, line 18, the goal of the work is to model multiple sources of uncertainty among different components; then why not authors have performed a simple ANOVA analysis, where you determine if there is any statistically significant difference b/n multiple independent groups and which component contribute more to projection uncertainty? Right now, a simple sensitivity analysis was performed to show DEM is the main contributor of uncertainty, however, whether the identified factor is statistically significant is yet not shown. Authors may refer (Gangrade et al., 2020) for this.

The suggested ANOVA analysis would have been possible if all the tested inputs were represented with a probability distribution, which is not the case for most of the inputs investigated (Table 1). We have referenced to this method in the discussion as a future research recommendation: "To avoid the computational burden of a Monte Carlo analysis, an ANOVA (Analysis of Variance) may be performed, as shown for example by Gangrade et al., (2020)", see page 12 line 15.

9. In page 12, Line 3-4 instead of cumbersome Monte Carlo based analyses a simple ANOVA may be performed to know which factor has significant contribution in overall uncertainty assessment.

This comment is similar to the previous one. See our addition to page 12 line 15.

10. In page 12, line 10-1, while the cited paper analysed compound flooding between storm surge and river discharge, (Ganguli and Merz, 2019) shows how compound flooding resulting from coastal total water level significantly impact the risk of inland riverine flooding over continental Europe.

The reference has been added at page 12, line 24.

We would also like to stress that the goal of the paper is to assess possible uncertainties in flood risk assessments as a result of uncertainties in the input data. Therefore, a (high) accuracy of the different models used (hydrodynamics, waves and damages) was not the primary objective of the paper, and we are confident that the approach used is suitable to address the initial research question.

A remark: P6 line 18. Do authors mean "WGS84"?

Indeed, the text has been updated.

References

[revised manuscript text omitted]

---

## Author Response (AR3)

**Reply letter manuscript "Uncertainties in Coastal Flood Risk Assessments in Small Island Developing States"**

Dear Dr. Parodi et al.

Thank you very much for the revised submission of your manuscript. I am happy with the revision. Before final acceptance, I have a minor comment.

I feel that your response to the following comment made by Reviewer 1 is not fully satisfactory:

7. In page 7, Line 21, the statement: "The choice of RCP 8.5 relies on the fact that the 90% confidence interval of this projection also captures most of the 90% confidence interval values under the RCP 4.5 scenario" This statement is not so clear, RCP 8.5 is altogether a different scenario when GHG emission reaches 8.5 W/m2 and considered to be most extreme; an overlap b/n two scenarios may misrepresent risk.

I would suggest to provide any evidence or cited literature in support of your statement: "The choice of RCP 8.5 relies on the fact that the 90% confidence interval of this projection also captures most of the 90% confidence interval values under the RCP 4.5 scenario"

Once you address the issue, I would be happy to accept the manuscript.

Best regards,

Animesh Gain

Dear Dr. Animesh Gain,

We thank the reviewer for the comment and apologize if this may have led to some misinterpretation on the use of the two scenarios. To address this comment, we have changed line 29 at page 7 accordingly: "The choice of RCP 8.5 relies on the fact that, for the area of interest, the 90% confidence interval of SLR projections under this scenario also captures the 50-90% percentiles of SLR projections under the RCP 4.5 scenario, i.e. it includes the more severe half of this milder scenario.". To clarify this, we have also included the figure below to the rebuttal which we hope clarifies this statement.

Kind regards,

Matteo Parodi

[Figure]

*Figure 1 90 percent confidence interval of Sea level rise projections through time from* (Vousdoukas et al., 2018), *under the RCP 4.5(blue shaded area) and 8.5 (red shaded area) scenarios for the area of study. The red and blue line indicate the 50th percentile.*

[revised manuscript text omitted]